# PrimeComposer: Faster Progressively Combined Diffusion for Image Composition with Attention Steering

Yibin Wang[*]
yibinwang1121@163.com
School of Computer Science,
Fudan University
Shanghai, China

Weizhong Zhang[*]
weizhongzhang@fudan.edu.cn
School of Data Science,
Fudan University
Shanghai, China

Jianwei Zheng
zjw@zjut.edu.cn
College of Computer Science and Technology,
Zhejiang University of Technology
Hangzhou, Zhejiang, China

Cheng Jin[†]
jc@fudan.edu.cn
School of Computer Science,
Fudan University
Shanghai, China

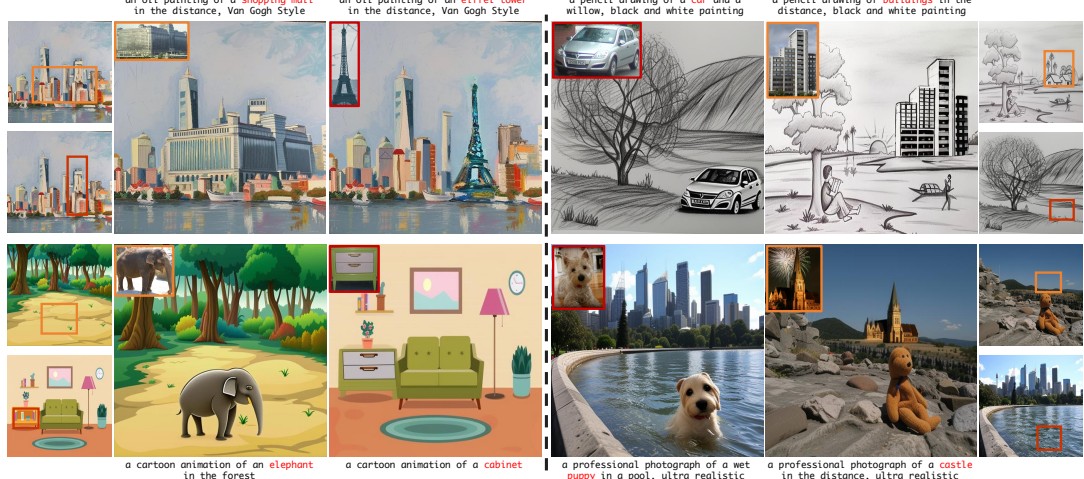

**Figure 1: Displayed are the results generated using PrimeComposer, showcasing its prowess across various domains: oil painting, sketching, cartoon animation, and photorealism.**

## ABSTRACT

Image composition involves seamlessly integrating given objects into a specific visual context. Current training-free methods rely on composing attention weights from several samplers to guide the generator. However, since these weights are derived from disparate contexts, their combination leads to coherence confusion and loss of appearance information. These issues worsen with their excessive focus on background generation, even when unnecessary in this task. This not only impedes their swift implementation but also compromises foreground generation quality. Moreover, these methods introduce unwanted artifacts in the transition area. In this paper, we formulate image composition as a subject-based local editing task, solely focusing on foreground generation. At each step, the edited foreground is combined with the noisy background to maintain scene consistency. To address the remaining issues, we propose PrimeComposer, a faster training-free diffuser that composites the images by well-designed attention steering across different noise levels. This steering is predominantly achieved by our Correlation Diffuser, utilizing its self-attention layers at each step. Within these layers, the synthesized subject interacts with both the referenced object and background, capturing intricate details and coherent relationships. This prior information is encoded into the attention weights, which are then integrated into the self-attention layers of the generator to guide the synthesis process. Besides, we introduce a Region-constrained Cross-Attention to confine the impact of

[*]Equal contribution.
[†]Corresponding Author.

specific subject-related words to desired regions, addressing the unwanted artifacts shown in the prior method thereby further improving the coherence in the transition area. Our method exhibits the fastest inference efficiency and extensive experiments demonstrate our superiority both qualitatively and quantitatively. The code is available at https://github.com/CodeGoat24/PrimeComposer.

## CCS CONCEPTS

• **Computing methodologies → Computer vision problems**.

## KEYWORDS

diffusion, image composition, image editing

## 1 INTRODUCTION

Image composition entails seamlessly incorporating the given object into the specific visual context without altering the object's appearance while ensuring natural transitions. Earlier studies employ personalized concept learning [14, 15, 20, 21, 35], yet they often rely on costly instance-based optimization and encounter limitations in generating concepts with specified backgrounds. Although these challenges are effectively addressed by utilizing diffusion models to explicitly incorporate additional guiding images [37, 46], retraining these pre-trained models on customized datasets risks compromising their rich prior knowledge. Consequently, these methods exhibit limited compositional abilities beyond their training domain, and they still demand substantial computational resources.

A recent study [28] develops a training-free framework, TF-ICON, that leverages attention weights from several samplers to composite the self-attention map of the diffuser during the composition process. Despite achieving notable success in this task, it still faces substantial challenges in preserving the appearance of complex objects (Fig. 2 (left)) and synthesizing a natural coherence (Fig. 2 (right)). The primary issue resides in its composite self-attention maps: the incorporation of attention weights from different contexts introduces potential ambiguity. To be precise, in TF-ICON, each sampler's weights are calculated within its specific global context [39], and the act of forcibly combining them leads to the synthesized confusion in coherent relations and loss of appearance information. This undoubtedly hampers its ability to accurately represent the intrinsic characteristics of the given object and establish a coherent relationship. These issues are further accentuated by its overemphasis on background generation (a tailored sampler). It is essential to underscore that the background inherently necessitates no alteration, given the user's exclusive focus on foreground area generation. This superfluous emphasis not only introduces computational overhead but also leads to compromise on the foreground synthesis. Moreover, this approach introduces the unwanted artifacts in the transition area as shown in Fig. 4, further hindering the natural coherence establishment.

In summary, while the current training-free method has mitigated the need for costly optimization and retraining, it remains incapable of capturing the nuanced appearance of objects and forging dependable coherent relations. Urgent exploration of more effective steering mechanisms for training-free composition, without compromising efficiency, is imperative.

In this paper, we novelly formulate this task as a subject-guided local editing problem, focusing solely on foreground generation due to the unnecessary alteration of the background. To achieve this, as depicted in Fig. 3, we utilize the pre-trained Latent Diffusion Model (LDM) [34] to edit local foreground areas evolvingly based on the given object and text. Then the resultant edited area is spatially combined with a certain noised background at each step to maintain the scene. To address the remaining issues, we propose a faster training-free method, dubbed PrimeComposer, a progressively combined diffusion model that integrates the user-provided object to the background through well-designed attention steering across different noise levels. This progressive steering is primarily facilitated by our Correlation Diffuser (CD). Specifically, in each step, we combine specific noisy-level versions of the provided object and background at the pixel level while simultaneously segmenting the synthesized subject from the previous step's results. These components are then fed into the CD's diffusion pipeline where the synthesized subject interacts with both the referenced object and background within self-attention layers. The interrelation information, encoded as prior weights, encapsulates rich mutual correlations and object appearance features. Consequently, we infuse them into LDM's self-attention maps (yellow and orange regions in Fig. 3 (bottom right), respectively) to meticulously steer the preservation of object appearance and ensure harmonious coherence establishment. To fortify the steering impact, we further advance the classifier-free guidance [17], elaborated in Sec. 4.5. Additionally, we introduce Region-constrained Cross-Attention (RCA), replacing cross-attention layers in LDM, to confine the impact of specific subject-related words to predefined regions in attention maps. This helps mitigate unwanted artifacts, thereby enhancing coherence in the transition area.

Note that CD is the only sampler for steering and all the infused attention weights are computed from the accordant context. However, intuitively infusing object appearance-related weights on all layers will result in the subject overfitting, as shown in Fig. 5, thereby leading to unexpected coherence problems: style inconsistency. Therefore, we propose to control appearance infusion *only in the decoder part of the U-Net* since the decoder has been proven to focus on learning the appearance and textures [52].

Our contributions can be summarized as follows. 1) We formulate image composition as a subject-guided local editing problem and propose a faster training-free method, namely, PrimeComposer. 2) We develop the CD to simultaneously alleviate the challenge of preserving complex objects' appearance and synthesizing natural coherence by well-designed attention steering. 3) We introduce RCA to address the unwanted artifacts in prior methods. 4) Our method exhibits the fastest inference efficiency and extensive experiments demonstrate our superiority both qualitatively and quantitatively.

## 2 RELATED WORK

Image composition serves as a valuable tool for diverse downstream tasks, e.g., entertainment, and data augmentation [6, 11, 25, 28, 42]. The practice for this task broadly falls into two categories: text-guided and image-guided composition.

Text-guided composition [1, 2, 7, 12, 26] involves generating images based on a text prompt that specifies multiple objects. This

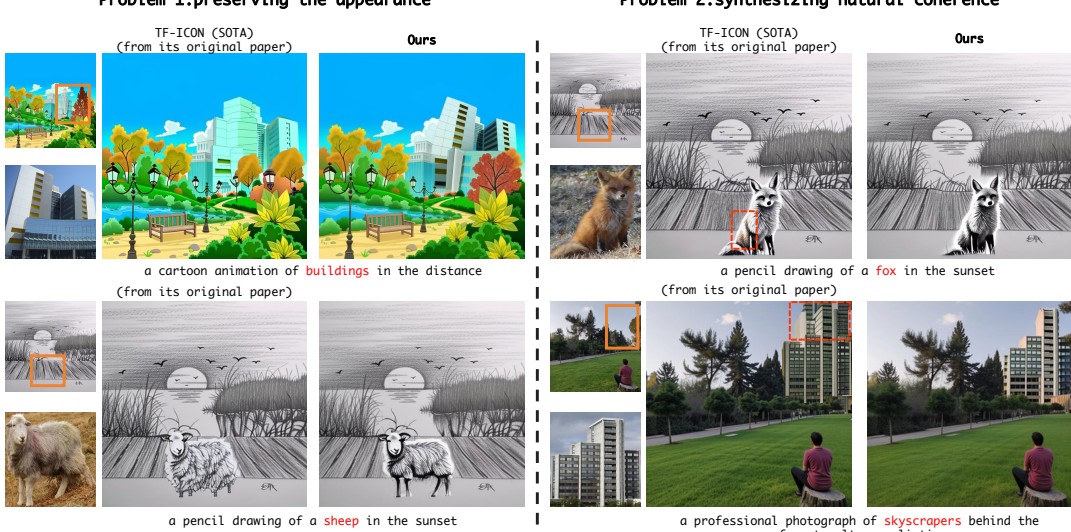

**Figure 2: Current methods encounter significant challenges in preserving the objects' appearance (left) and synthesizing natural coherence (right). The problematic areas of coherence are indicated by red dotted lines.**

approach allows for diverse appearances as long as the semantics align with the prompt. Despite its effectiveness, semantic errors may arise, especially with prompts involving multiple objects. These errors, including attribute leakage and missing objects, often necessitate extensive prompt engineering [43].

Conversely, image-guided composition [4, 13, 22, 28, 37, 45–47] incorporates specific objects and scenarios from user-provided photos, potentially with the assistance of a text prompt. This approach presents greater challenges, particularly when dealing with images from different visual domains. Specifically, image-guided composition encompasses various sub-tasks [31], such as object placement [3, 8, 23, 38, 40, 41, 48], image blending [44, 49], image harmonization [9, 10, 19, 30, 32, 45], and shadow generation [18, 24, 36, 51]. These diverse tasks are typically tackled by distinct models and pipelines, showcasing the intricacy of image-guided composition. Recently, diffusion models have demonstrated impressive capabilities in image-guided composition by simultaneously tackling all these subtasks. While prior studies have explored personalized concept learning [14, 15, 20, 21, 35], they often rely on costly instance-based optimization and face limitations in generating concepts with specified backgrounds. To overcome these challenges, subsequent studies [37, 46] effectively incorporate additional guiding images into diffusion models through retraining pre-trained models on tailored datasets. However, this poses a risk of compromising their rich prior knowledge. Besides, these models exhibit limited compositional abilities beyond their training domain and demand substantial computational resources. A recent study [28] introduces a training-free method involving the gradual injection of composite self-attention maps through multiple samplers. Despite its remarkable success, it encounters challenges in preserving the appearance of complex objects and synthesizing natural coherence.

Diverging from the approaches mentioned earlier, we novelly formulate this task as an subject-guided local editing problem. Our progressively combined Diffusion, PrimeComposer, intricately depicts the object and achieves harmonious coherence through well-designed attention steering across a progression of noise levels.

## 3 PRELIMINARY

Denoising diffusion probabilistic models (DDPMs) [16] are designed to reverse a parameterized Markovian image noising process. They start with isotropic Gaussian noise samples and gradually transform them into samples from a training distribution by iteratively removing noise. Given a data distribution $\boldsymbol{x}_0 \sim q(\boldsymbol{x}_0)$, the forward noising process produces a sequence of latents $\boldsymbol{x}_1, ..., \boldsymbol{x}_T$ by adding Gaussian noise with variance $\beta_t \in (0, 1)$ at each time step $t$:

$$q(\boldsymbol{x}_1, ..., \boldsymbol{x}_T | \boldsymbol{x}_0) = \prod_{t=1}^{T} q(\boldsymbol{x}_t | \boldsymbol{x}_{t-1}), \tag{1}$$

$$q(\boldsymbol{x}_t | \boldsymbol{x}_{t-1}) = \mathcal{N}(\sqrt{1 - \beta_t} \boldsymbol{x}_{t-1}, \beta_t \mathbf{I}).$$

When $T$ is sufficiently large, the last latent $\boldsymbol{x}_T$ approximates an isotropic Gaussian distribution.

An important property of the forward noising process is that any step $\boldsymbol{x}_t$ can be directly sampled from $\boldsymbol{x}_0$, without generating the intermediate steps:

$$q(\boldsymbol{x}_t | \boldsymbol{x}_0) = \mathcal{N}(\sqrt{\bar{\alpha}_t} \boldsymbol{x}_0, (1 - \bar{\alpha}_t) \mathbf{I}), \tag{2}$$

$$\boldsymbol{x}_t = \sqrt{\bar{\alpha}_t} \boldsymbol{x}_0 + \sqrt{1 - \bar{\alpha}_t} \boldsymbol{\epsilon},$$

where $\boldsymbol{\epsilon} \sim \mathcal{N}(0, \mathbf{I})$, $\alpha_t = 1 - \beta_t$, and $\bar{\alpha}_t = \prod_{s=0}^{t} \alpha_s$.

To draw a new sample from the distribution $q(\boldsymbol{x}_0)$, the Markovian process is reversed. Starting from a Gaussian noise sample $\boldsymbol{x}_T \sim \mathcal{N}(0, \mathbf{I})$ with $\alpha_t = 1 - \beta_t$, a reverse sequence is generated by sampling the posteriors $q(\boldsymbol{x}_{t-1} | \boldsymbol{x}_t, \boldsymbol{x}_0)$.

However, $q(\boldsymbol{x}_{t-1} | \boldsymbol{x}_t, \boldsymbol{x}_0)$ is unknown and depends on the unknown data distribution $q(\boldsymbol{x}_0)$. To approximate this function, a

deep neural network $p_\theta$ is trained to predict the mean and covariance of $x_{t-1}$ given $x_t$ as input:

$$p_\theta(x_{t-1}|x_t) = \mathcal{N}(\mu_\theta(x_t, t), \Sigma_\theta(x_t, t)). \tag{3}$$

Rather than inferring $\mu(x_t, t)$ directly, Ho et al. [16] propose to predict the noise $\epsilon_\theta(x_t, t)$ that was added to $x_0$ to obtain $x_t$ according to Equation 2. Then, $\mu(x_t, t)$ is derived using Bayes' theorem:

$$\mu_\theta(x_t, t) = \frac{1}{\sqrt{\alpha_t}} \left( x_t - \frac{\beta_t}{\sqrt{1 - \bar{\alpha}_t}} \epsilon_\theta(x_t, t) \right). \tag{4}$$

For more detail see [16]. In this work, we leverage the pre-trained text-to-image Latent Diffusion Model (LDM) [34], a.k.a. Stable Diffusion, which applies the noising process in the latent space.

# 4 METHOD

This section begins with an overview of our method, followed by an in-depth explanation of self-attention steering based on our Correlation Diffuser (CD). Subsequently, we will explore the details of our Region-constrained Cross-Attention (RCA). Finally, we will introduce our careful extension of classifier-free guidance (CFG) during inference.

## 4.1 Overview

This work formulates image composition as a local object-guided editing task, utilizing LDM to depict the object and synthesize natural coherence. We aim to seamlessly synthesize the given object within specific foreground areas effectively without compromising efficiency. To achieve this, we propose PrimeComposer, a faster training-free progressively combined composer that composites the images by well-designed attention steering across different noise levels. Specifically, we leverage the prior attention weights from CD to steer the preservation of object appearance and the establishment of natural coherent relations. Its effectiveness is further enhanced through our extension of CFG. Besides, we introduce RCA to replace the cross-attention layers in LDM. RCA effectively restricts the influence of object-specific words to desired spatial regions, thereby mitigating unexpected artifacts around synthesized objects.

In the depicted pipeline (Fig. 3), the process begins with a background image, an object image, a caption prompt $P$, and two binary masks $M^{obj}$ and $M^{fg}$ (designating the object and foreground areas, respectively). The background and object images are first inverted into latent representations $z_T^{bg*}$ and $z_T^{obj*}$ using DPM-Solver++ [27] following [28]. These representations are then composited at the pixel level based on $M^{obj}$. To harness the prior knowledge of LDM for synthesizing the coherence, Gaussian noise is introduced to the transition areas (where $M^{obj}$ XOR $M^{fg}$), resulting in the initial input noise $z^{init}$. In each step $t$, we discern the attention weights embodying object appearance features and coherent correlations from CD's self-attention layers. These prior weights are then infused into LDM's self-attention maps to guide foreground generation. Additionally, all cross-attention maps in LDM are rectified to limit the impact of object-specific words in predefined regions. To preserve the unchanged scene, the edited foreground areas at each step are combined with the certain noisy version of the background $z_{t-1}^{bg*}$ based on $M^{fg}$. This iterative process ensures seamless composition.

## 4.2 Self-Attention Steering

While the composite noise $z^{init}$ acts as the initial input, and the caption prompt $P$ contributes to inpainting the transition areas, LDM still encounters challenges in preserving the appearance of the object and synthesizing harmonious results effectively as shown in Fig. 7. To tackle this, we propose the CD to chase down the attention weights that encapsulate rich prior semantic information of the object's features and coherent relations. Subsequently, these prior attention weights are employed to guide the synthesis process in the initial $\alpha T$ steps where $\alpha$ is the hyperparameter.

*4.2.1 Correlation diffuser.* The CD is adapted from the pre-trained Stable Diffusion with tailored self-attention layers to generate prior attention maps. Specifically, at each timestep $t$, it takes the pixel composite image $z_t^{pc*}$ (derived from the specific noise version of the user-provided object and background) and the latent representation of the synthesized object $z_{t,obj}$ (segmented from the previous step's result $z_t$) as input. In each self-attention layer $l$, the self-attention map $A_{l,t}$ are computed as follows:

$$q_{l,t} = z_t^{pc*,l} W_l^q, \quad k_{l,t} = z_{t,obj}^l W_l^k, \tag{5}$$

$$A_{l,t} = \text{Softmax}\left( q_{l,t} \cdot (k_{l,t})^T / \sqrt{d} \right), \tag{6}$$

where $A_{l,t} \in \mathbb{R}^{(h \times w) \times n}$, $h$, $w$ denote the height and width of background, $n$ denote the flatten pixel amount of the object and $W_l^q$, $W_l^k$ are projection matrices.

*4.2.2 Prior weights infusion.* The obtained prior attention map $A_{l,t}$ comprises two constituents: $A_{l,t}^{cross} \in \mathbb{R}^{(h \times w - n) \times n}$ and $A_{l,t}^{obj} \in \mathbb{R}^{n \times n}$. $A_{l,t}^{cross}$ reflects the relations between the synthesized object and background, while $A_{l,t}^{obj}$ contains the object appearance features. These constituents are then infused into the $l$-th self-attention maps $A_{l,t}^{ldm}$ in LDM, as illustrated by the yellow and orange regions in Fig. 3 (bottom right), respectively. The process can be formulated as: $A_{l,t}^* = \vartheta_{\text{infusion}}(A_{l,t}^{cross}, A_{l,t}^{obj}, A_{l,t}^{ldm})$, where $\vartheta_{\text{infusion}}$ is the function to produce the infused self-attentionmaps $A_{l,t}^*$.

However, the intuitive infusion of $A_{l,t}^{obj}$ on all layers may result in the synthesized object closely resembling the given image, i.e., subject overfitting. This, in turn, can lead to unexpected coherence problems, as depicted in Fig. 5. To address this issue, we propose a controlled approach to appearance infusion: restricting it only to *the decoder part of the U-Net*. This decision is grounded in the understanding that the decoder primarily focuses on learning appearance and texture [52], thus promoting more natural coherence in the synthesized output.

## 4.3 Region-constrained Cross Attention

Following [28], the caption prompt $P$ is utilized to guide the synthesis of transition areas. However, this introduces a coherence problem, as demonstrated in Fig. 4 and Fig. 7: the newly generated object isn't consistently guaranteed to appear appropriately within the regions outlined by $M^{obj}$, thereby causing the unwanted artifacts. To address this challenge, we introduce RCA, which replaces the cross-attention layers in the U-Net to restrict the impact of object-specific words to regions defined by $M^{obj}$.

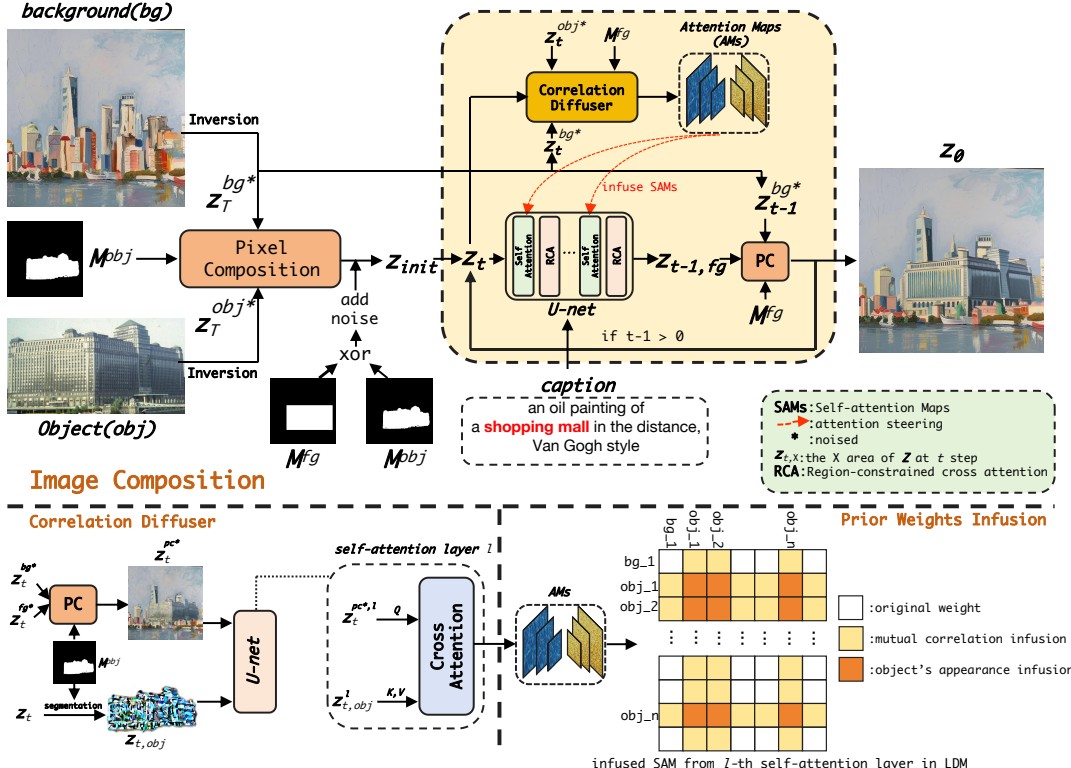

**Figure 3: The overview of our PrimeComposer.**

Specifically, the latent noisy image is projected to a query matrix $q$, while the text prompt's embedding is projected to the key $k$ and value $v$ matrices via learned linear projections. The cross-attention maps $A \in \mathbb{R}^{(h \times w) \times p}$ is computed as $A = q \cdot k^{\mathrm{T}} / \sqrt{d}$, where $p$ denote the amount of tokens.

Then, certain attention maps $A^{obj}$ corresponding to object-related tokens (e.g., 'white fox' and 'lemon' in Fig. 4) in $A$ are rectified by applying the binary mask $M^{obj}$:

$$A^{obj} = \begin{cases} a_{ij}^{obj}, & m_{ij} = 1, \\ -inf, & m_{ij} = 0, \end{cases} \qquad (7)$$

where $a_{ij}^{obj}$ and $m_{ij}$ represents the weight of the object-related token's map and the value of $M^{obj}$, respectively, at the position $(i, j)$. After that, we obtain the rectified attention maps $\hat{A}$ and the output of RCA layer is defined as $\mathcal{F} = \mathrm{Softmax}(\hat{A})v$.

Through the use of rectified attention maps, this module effectively restricts the impact of object-related words to specific spatial regions on the image features. Consequently, the model can enforce the generation of objects in desired positions and shapes, addressing the coherence problem highlighted earlier.

### 4.4 Background Preserving Combining

Inspired by the concept of blending two images by separately combining each level of their Laplacian pyramids [5], our approach involves combining synthesized foreground areas and the given background across different noise levels to maintain the unchanged scene. The underlying principle is that at each step in the diffusion process, a noisy latent is projected onto a manifold of naturally noised images at a specific level. While blending two noisy images (from the same level) may result in output likely outside the manifold, the subsequent diffusion step projects the result onto the next-level manifold, thereby improving coherence [2].

Thus, at each step $t$, starting from a latent $z_t$, we perform a single diffusion step. The resultant latent is segmented based on $M^{fg}$, yielding a latent denoted $z_{t-1,fg}$. In addition, we obtain a noised version of the input background $z_{t-1}^{bg*}$ using Equ. 2. The two latents are combined using the foreground mask:

$$z_{t-1} = z_{t-1,fg} \odot M^{fg} + z_{t-1}^{bg*} \odot (1 - M^{fg}), \qquad (8)$$

### 4.5 Extended Classifier-free Guidance

To reinforce the steering effect of the infused prior weights in foreground generation, CFG is extended in each sampling step to extrapolate the predicted noise $\hat{\epsilon}$ along the direction specified by certain infusions:

$$\hat{\epsilon} = \epsilon_\theta(z_t | \varnothing) + s \left[ \epsilon_\theta(z_t | c, f) - \epsilon_\theta(z_t | f) + \epsilon_\theta(z_t | c, f) - \epsilon_\theta(z_t | c) \right].$$

Where $\varnothing$, $c$, and $f$ signify a null, caption prompt, and infusion condition, respectively. $\epsilon_\theta$ and $\hat{\epsilon}$ represent the employed LDM and its output noise. The hyperparameter $s > 0$ denotes the guidance scale, and the reinforcement effect becomes stronger as $s$ increases.

As shown in Fig. 7, this careful design effectively strengthens the ability to generate more harmonious images, since this leads

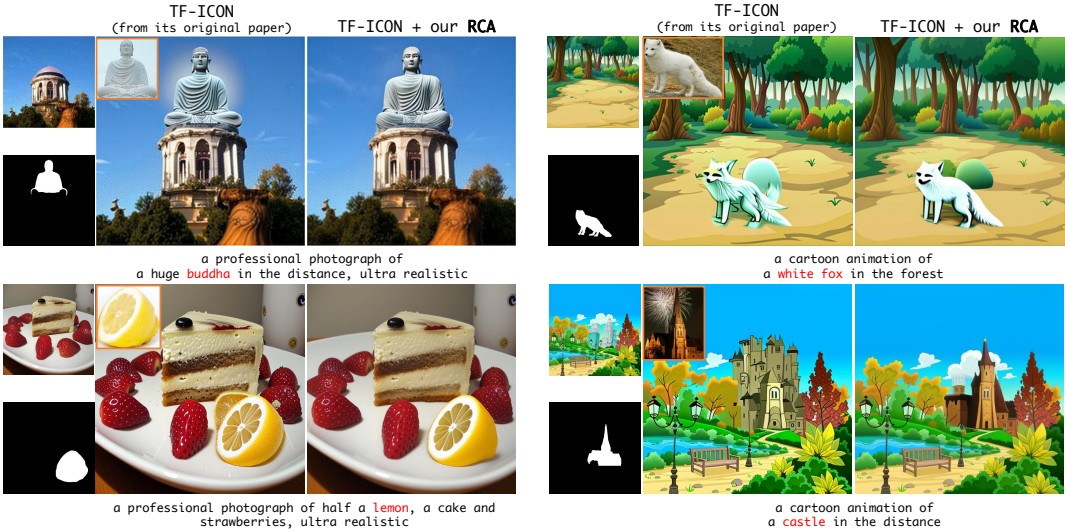

**Figure 4: The effectiveness of our Region-constrained Cross Attention.**

LDM to become more adept at capturing and preserving the subtle details of the object's appearance and coherence relations. The visualization of the saliency maps derived from our extended CFG is provided in the supplementary.

## 5 EXPERIMENTS

### 5.1 Implementation Details

*5.1.1 Test benchmark.* We employ the only publicly released cross-domain composition benchmark [28], which contains 332 samples. Each sample consists of a background image, an object image, a foreground mask, an object mask, and a caption prompt. The background images comprise four visual domains: photorealism, pencil sketching, oil painting, and cartoon animation. We adjust all the caption prompts to mark the object-specific words. The details are left to supplementary.

*5.1.2 Baselines.* We conduct a qualitative comparison between our method and state-of-the-art baselines. The baselines includes Deep Image Blending (DIB) [49], Blended Diffusion [2], Paint by Example [46], SDEdit [29], and TF-ICON [28]. For the quantitative assessment, all baselines are considered, excluding DCCF, as it is designed for harmonizing images after copy-and-paste operations.

*5.1.3 Test configures.* Given that most baselines are trained primarily in the photorealism domain, where objective metrics are more effective, we conducted our quantitative comparison specifically in this domain for the fairness. For other domains, we relied on a user study and qualitative comparisons. We utilized the official implementation of all the baselines. Our framework employed the pre-trained Stable Diffusion with the second-order DPM-Solver++ in 20 steps. The hyperparameter $\alpha$ for prior weights infusion was set to 0.2 and the scale of classifier-free guidance was set to 2.5 for the photorealism domain and 5 for other cross-domains.

*5.1.4 Evaluation metrics.* We evaluate our method using four metrics: (1) $LPIPS_{(BG)}$: measures background consistency based on the

| Methods | $LPIPS_{(BG)}$ ↓ | $LPIPS_{(FG)}$ ↓ | $CLIP_{(Image)}$ ↑ | $CLIP_{(Text)}$ ↑ |
|---|---|---|---|---|
| WACV'20 DIB[49] | 0.11 | 0.63 | 77.57 | 26.84 |
| ICLR'22 SDEdit[29] | 0.42 | 0.66 | 77.68 | 27.98 |
| CVPR'22 Blended[2] | 0.11 | 0.77 | 73.25 | 25.19 |
| CVPR'23 Paint[46] | 0.13 | 0.73 | 80.26 | 25.92 |
| ICCV'23 TF-ICON[28] | 0.10 | 0.60 | 82.86 | 28.11 |
| **Ours** | **0.08** | **0.48** | **84.71** | **30.26** |

**Table 1: Quantitative evaluation results for image composition in the photorealism domain.**

LPIPS metric [50]. (2) $LPIPS_{(FG)}$: evaluates the low-level similarity between the object region and the reference using the LPIPS metric [50]. (3) $CLIP_{(Image)}$: assesses the semantic similarity between the object region and the reference in the CLIP embedding space [49]. (4) $CLIP_{(Text)}$: measures the semantic alignment between the text prompt and the resultant image [33].

### 5.2 Qualitative Comparisons

The qualitative comparison results in Fig. 6 highlight the superior performance of PrimeComposer in seamlessly integrating objects across various domains while preserving their appearance and synthesizing natural coherence. Notably, some baselines, such as Paint by Example, face challenges in maintaining the appearance of the given object. Blended Diffusion, relying solely on text prompts, fails to ensure the synthesized object matches the reference image. Additionally, Deep Image Blending, DCCF and SDEdit struggle to achieve harmonious transitions. While TF-ICON performs relatively well compared to other methods, it still faces difficulties in simultaneously preserving object appearance and synthesizing natural coherence. For example, it fails to preserve the identity features of the tortoise in the first row and struggles to achieve optimal coherence around the synthesized building in the last row. Additional qualitative comparison results are left to supplementary.

### 5.3 Quantitative Analysis

As demonstrated in Tab. 1, our proposed PrimeComposer outperforms all competitors consistently across all metrics, highlighting the exceptional visual quality of the composite images it generates.

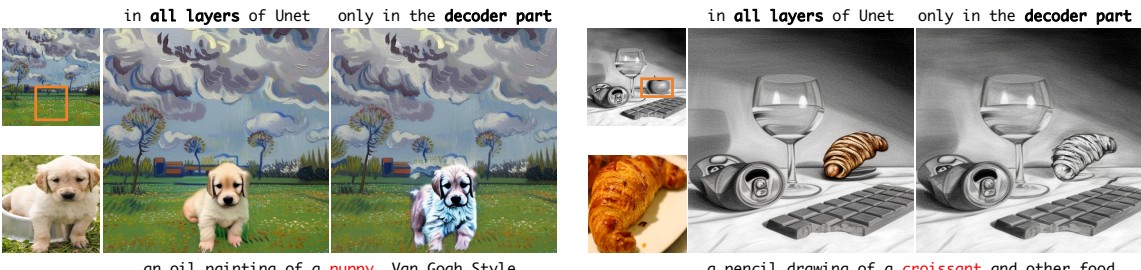

**Figure 5: Qualitative results regarding the unexpected coherence problem, i.e., style inconsistency.**

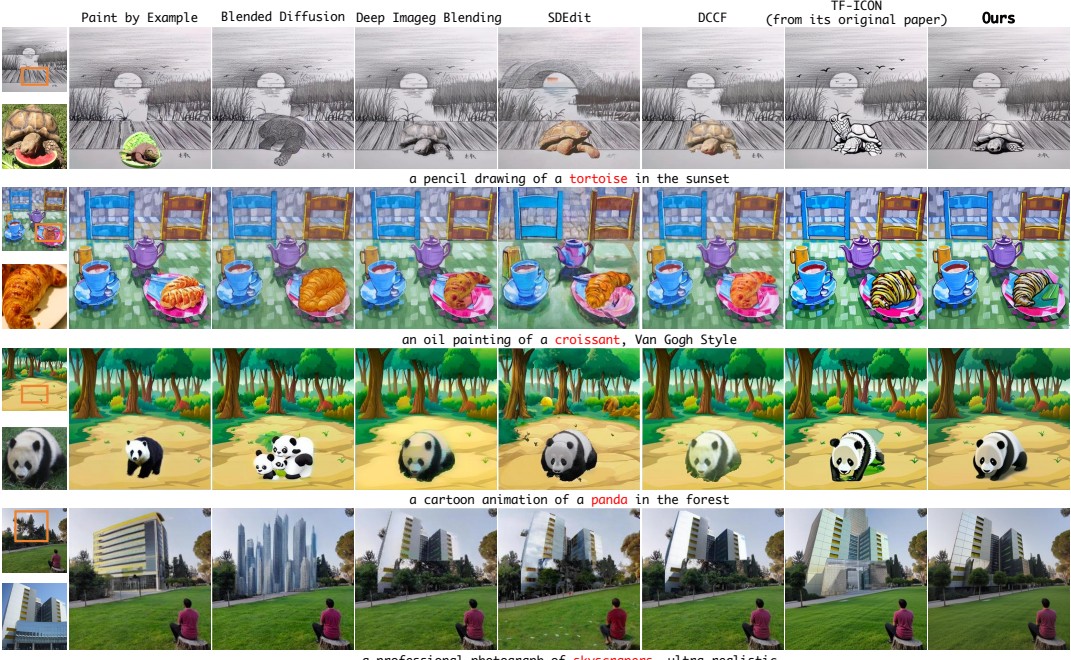

**Figure 6: Qualitative comparison with SOTA baselines in cross-domain image composition. All the results of TF-ICON come from its originary paper.**

Notably, TF-ICON also exhibits commendable generation quality. However, its composite self-attention map, derived from multiple samplers, introduces synthesis confusion. As a consequence, the realism and harmony of its resulting images are compromised. In contrast, our PrimeComposer Competently alleviates these challenges. Notably, PrimeComposer surpasses it by 1.85 and 2.15 on $CLIP_{(Image)}$ and $CLIP_{(Text)}$, respectively. This demonstrates that our method outperforms previous approaches in preserving object appearance and achieving harmonious coherence synthesis.

## 5.4 Inference Time Comparison

We conduct a comparison with the previous SOTA training-free method on an NVIDIA A100 40GB PCIe. Considering the time depends on the size of the user mask and reference image, we measure the averaged inference time per image across various

domains in the test benchmark to ensure fairness. As shown in Tab. 2, PrimeComposer consistently exhibits faster inference times than TF-ICON across all domains. Remarkably, our inference time in the photorealism domain is notably lower than TF-ICON, specifically achieving a time of 14.32 seconds. This phenomenon is expected, considering that TF-ICON employs four samplers for composition, while our approach only utilizes two samplers, i.e., the pre-trained LDM and a Correlation Diffuser, during the process, resulting in a more efficient computational performance. See supplementary for additional inference speed comparisons with training baselines, which further proves the efficiency of our method.

## 5.5 User Study

We conduct a user study to compare image composition baselines across different domains. Specifically, we invited 30 participants

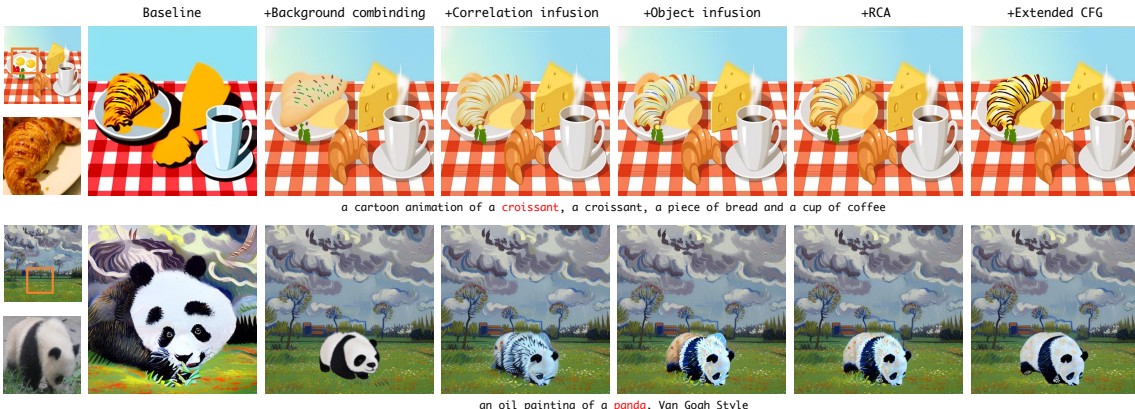

**Figure 7: Ablation study of different variants of our framework. RCA: Region-constrained Cross-Attention. CFG: Classifier-free Guidance.**

| Methods | PI in *Cartoon* | PI in *Sketch* | PI in *Painting* | PI in *Photorealism* |
|---|---|---|---|---|
| TF-ICON[28] | 28.98 sec | 29.12 sec | 29.75 sec | 30.55 sec |
| **Ours** | **16.62** sec | **15.25** sec | **15.58** sec | **16.23** sec |

**Table 2: Inference time comparison with the previous SOTA training-free method on various domains. PI and sec mean Per Image and seconds, respectively.**

through voluntary participation and assigned them the task of completing 40 ranking questions. The ranking criteria comprehensively considered factors including foreground preservation, background consistency, seamless composition, and text alignment. The results are presented in Tab. 4, where the domain information is formatted as 'foreground domain & background domain'. Notably, our method received favorable feedback from the majority of participants across various domains.

## 5.6 Ablation Study

we conduct ablations on key design choices in the following cases: (1) Baseline, where the composition is synthesized by LDM from $T$ to 0 only with the caption prompt. The initial noise is the pixel composition derived from inverted codes of the given object and background; (2) Background combining is applied to maintain the unchanged scene; (3) Correlation infusion is employed to steer the natural coherent relation establishment; (4) Object infusion is employed to steer the preservation of the object's appearance; (5) Region-constrained Cross-Attention is used to enforce the generation of objects in desired positions and shapes, addressing the coherence problem caused by the caption prompt; (6) Extended CFG, tailored to reinforce the steering impact of prior weights infusion.

**Quantitative results**: Tab. 3 presents quantitative ablation results, showcasing the superior performance of our complete algorithm across all metrics except $CLIP_{(Text)}$. It is noteworthy that the baseline achieves the best $CLIP_{(Text)}$ result, as it generates compositions solely relying on the caption prompt without any additional constraints [28]. These ablation results underline the effectiveness of our proposed algorithm in enhancing various aspects of the composition process.

| Methods | $LPIPS_{(BG)}$ ↓ | $LPIPS_{(FG)}$ ↓ | $CLIP_{(Image)}$ ↑ | $CLIP_{(Text)}$ ↑ |
|---|---|---|---|---|
| Baseline | 0.40 | 0.56 | 74.68 | **31.32** |
| +Background combining | 0.10 | 0.53 | 75.49 | 30.01 |
| +Correlation infusion | 0.09 | 0.51 | 82.51 | 29.87 |
| +Object infusion | 0.09 | 0.50 | 83.36 | 30.18 |
| +RCA | 0.08 | 0.48 | 84.12 | 30.24 |
| +Extended CFG | **0.08** | **0.48** | **84.71** | 30.26 |

**Table 3: Ablation study: quantitative comparison of various variants of our framework.**

| Methods | P & P | P & O | P & S | P & C | Total |
|---|---|---|---|---|---|
| Blended[29] | 2.14 | 1.55 | 1.78 | 2.54 | 1.86 |
| SDEdit[2] | 3.09 | 2.88 | 2.43 | 2.97 | 2.71 |
| Paint[46] | 3.31 | 2.93 | 2.13 | 2.86 | 2.85 |
| DCCF[49] | 3.76 | 3.35 | 3.01 | 3.58 | 3.44 |
| TF-ICON[28] | 4.23 | 4.46 | 3.92 | 4.39 | 4.31 |
| **Ours** | **4.36** | **4.49** | **4.43** | **4.52** | **4.44** |

**Table 4: User study: higher score, better ranking. P: photorealism; O: oil painting; S: sketchy painting; C: cartoon.**

**Qualitative results**: To further visualize the effectiveness of each design choice, we provide qualitative results shown in Fig. 7. These results directly prove the indispensable role of all design choices. More qualitative ablation results are left to supplementary.

## 6 CONCLUSION

In this paper, we formulate image composition as a subject-guided local image editing task and propose a faster training-free diffuser, PrimeComposer. Leveraging well-designed attention steering, primarily through the Correlation Diffuser, our method seamlessly integrates foreground objects into noisy backgrounds while maintaining scene consistency. The introduction of Region-constrained Cross-Attention further enhances coherence and addresses unwanted artifacts in prior methods. Our approach demonstrates fastest inference efficiency and outperforms existing methods both qualitatively and quantitatively in extensive experiments.

## 7 ACKNOWLEDGMENTS

This work was supported by National Natural Science Fund of China (62176064) and Shanghai Municipal Science and Technology Commission (22dz1204900).

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
