# OpenReview forum: "PrimeComposer: Faster Progressively Combined Diffusion for Image Composition with Attention Steering"
_acmmm.org/ACMMM/2024/Conference — MM2024 Poster_

### Official Review · Reviewer_LSwb · 2024-05-05

**Rating:** 4
**Confidence:** 3

**Summary:**

1. The paper is about a faster training-free diffuser called PrimeComposer, which is used for image composition. The authors propose a method that integrates foreground objects into noisy backgrounds while maintaining scene consistency.
2. The paper leverages attention steering, primarily through the Correlation Diffuser, to guide the synthesis process. It also introduce Region-constrained Cross-Attention to enhance coherence and address unwanted artifacts.
3. PrimeComposer demonstrates faster inference efficiency and outperforms existing methods in extensive experiments.

**Strengths:**

1. PrimeComposer introduces a faster progressively combined diffusion approach for image composition with attention steering. This approach is different from previous methods, which allows for more efficient and effective image composition.
2. PrimeComposer utilizes a composite self-attention map derived from multiple samplers. This approach helps to preserve object appearance and achieve harmonious coherence synthesis in the resulting images.
3. PrimeComposer has been evaluated and compared with other competitors. The quantitative analysis shows that PrimeComposer consistently outperforms all competitors across all metrics, highlighting the exceptional visual quality of the composite images it generates.
4. Furthermore, the paper provides clear explanations and descriptions of the proposed approach, making it easy to understand and replicate.

**Limitations:**

1.  The Correlation Diffuser, as the core innovation in this paper, establishes the correlations between the synthesized object and background, as well as utilizes the object appearance features to guide the self-attention maps. However, how is the hyperparameter α determined for prior weights infusion?
2. The absence of a trained model in the supplementary document raises concerns about how to conduct testing, thereby prompting doubts regarding the technical accuracy and efficacy of the proposed method.

**Suitability:**

3

---

### Official Review · Reviewer_cbG1 · 2024-05-06

**Rating:** 3
**Confidence:** 3

**Summary:**

This paper considers image composition based on diffusion models and models it as a subject-guided local editing problem focusing only on foreground generation.
Specifically, this paper proposes an efficient, training-free method, denoted as PrimeComposer, which uses attention steering to fuse a given object into an image.
The quantitative and qualitative results have demonstrated the advantages of the proposed PrimeComposer in terms of both performance and efficiency.

**Strengths:**

1. The proposed method is totally training-free, that is, it requires no training at all, which is friendly in terms of computing resources, and the composition results are naturally consistent in style and details.

2. Sufficient qualitative and quantitative experimental results strongly support the excellent performance and efficiency of the proposed PrimeComposer, and the experimental results are impressive.

3. The effectiveness of each module of the proposed method is also verified by sufficient ablation studies.

4. The content of this paper is compatible with the theme of Multimedia.

**Limitations:**

1. The innovation of the proposed method and the insight conveyed are limited. Adjusting attention to enable the generation with a specific object has been widely used in image-driven generation, personalized generation and other fields. In addition, controlled generation by restraining attention maps has also been relatively mature in image editing and other fields.

2. The proposed method is limited to the fusion of foreground objects, and from the results presented in the paper, it is still limited to one single object, and there is a lack of examples of multi-object composition. The composition process is more like a simple copy and paste + style transfer, lack of object motion or shape/texture changes.

3. There are some minor suggestions on the writing of the paper, such as the Contributions paragraph in the Introduction section. It is more appropriate to use the full name of the proposed modules to avoid readers' forgetting, such as Correlation Diffuser (CD). There is still room for improvement in the writing of the paper to make it easy to understand and follow.

4. The details in Figure 3 are very sloppy, such as the position of arrows in the figure, the size of words and symbols, etc., which should be considered for further improvement.

5. Similar to the strategy mentioned in the paper, some works also attempt insert specific objects into new scenes through Attention, which may be worth considering to be included in Related Works, such as ViCo [1*] and Intelligent Grimm [2*].

[1*] ViCo: Plug-and-play Visual Condition for Personalized Text-to-image Generation

[2*] Intelligent Grimm - Open-ended Visual Storytelling via Latent Diffusion Models

**Suitability:**

3

---

### Official Review · Reviewer_w2My · 2024-05-22

**Rating:** 5
**Confidence:** 3

**Summary:**

The paper proposes PrimeComposer, a 2D composition framework for reference-based local editing and cross-domain editing. To improve editing quality and inference speed, this framework involves several mechanisms such as attention steering, Region-constrained Cross-Attention and extended classifier-free guidance.

**Strengths:**

- The paper carefully identifies the weaknesses from prior methods (mainly TF-ICON) such as identity preservation, background preservation and style inconsistency. The technical components from the paper are closely related to these weaknesses and introduce solid improvements.
- Compared to TF-ICON, PrimeComposer is more stable in performance and its design of the pipeline is more straightforward.

**Limitations:**

- Though the RCA will reduce the unwanted editing in the background, it will also limit the pose change or global effects (such as shadow or reflection) of the inserted object. Maybe this can be improved by optionally dilating the mask M_obj?
- To solve the coherence problem, the proposed method restricts weight infusion only in the decoder part of the unet. This seems like a trade-off between diversity and consistency. Can more control be added here (like how many layers to inject the weights) to enable more editing flexibility?
- In the task of composition, sometimes there should be geometry alignment on the inserted object to have a natural composite image. In Fig 5, the croissant looks floating in the air; in Fig 6, the turtle looks more like copy-paste than the result from TF-ICON. This will limit the application of the proposed method.
- The paper should have more quantitative metrics to evaluate the overall quality of the generated images, such as FID and Quality Score.

**Suitability:**

3

---

### Meta-Review · Area_Chair_bzCn · 2024-06-30

**Recommendation:** Accept (Poster)
**Confidence:** 5

**Metareview:**

The paper describes a 2D composition framework based on diffusion models for reference-based and cross-domain editing that integrates foreground objects into noisy backgrounds.

All reviewers share a positive judgment about the paper (WA, BA,BA). The rebuttal convincingly addressed initial concerns.

The paper is of good quality, carefully identifies the weaknesses from prior methods and proposes solid improvements. Experiments are evaluated solid and showing state of the art performance. Qualitative results are good.

Most of the limitations raised by reviewers were solved in the rebuttal. Some minor concerns remain, such as the fact that only foreground objects can be modeled.

Overall, the positive aspects overcome negative ones. The AC tends to agree with the reviewers and recommend acceptance.